# Exploring Strategies to Promote Exercise as a Viable Obesity and Chronic Disease Treatment

**DOI:** 10.3390/nu17121997

**Published:** 2025-06-13

**Authors:** Kyle D. Flack, Matthew A. Stults-Kolehmainen, Robert E. Anderson, Reed Handlery, Seth A. Creasy, Victoria A. Catenacci

**Affiliations:** 1Research Institute Health and Wellness Center, Arkansas Colleges of Health Education, Fort Smith, AR 72916, USA; 2Yale-New Haven Hospital, New Haven, CT 06510, USA; matthew.stults@ynhh.org; 3Department of Biobehavioral Sciences, Teachers College, Columbia University, New York, NY 10027, USA; 4Gillings School of Global Public Health, University of North Carolina at Chapel Hill, Chapel Hill, NC 27514, USA; robert.anderson@unc.edu; 5School of Physical Therapy, Arkansas Colleges of Health Education, Fort Smith, AR 72916, USA; reed.handlery@achehealth.edu; 6Division of Endocrinology, Metabolism, and Diabetes, University of Colorado Anschutz Medical Campus, Aurora, CO 80045, USA; seth.creasy@cuanschutz.edu (S.A.C.); vicki.catenacci@cuanschutz.edu (V.A.C.); 7Anschutz Health and Wellness Center, University of Colorado Anschutz Medical Campus, Aurora, CO 80045, USA

**Keywords:** obesity, weight loss, exercise, adoption, chronic diseases

## Abstract

Obesity and its related comorbidities continue to be a primary public health concern, especially in the United States (US). Such comorbidities include the top two causes of death in the US: cardiovascular disease and cancer. Obesity is also associated with several other chronic conditions that affect millions of adults and children, including diabetes, kidney, and liver disease. Weight loss has long been considered the front-line treatment and prevention strategy for these conditions. Lifestyle approaches, including dietary modification and increasing physical activity, are typically recommended for individuals with obesity, although rates of achieving and maintaining clinically meaningful weight loss remain low. Understanding the root causes of minimal weight loss and weight regain has been a prime focus among many researchers over the past several decades. The present review addresses several advantages of prioritizing exercise as an obesity and chronic disease treatment. We discuss current challenges when exercise is the primary treatment strategy, including physiological parameters that may influence the efficacy of exercise in addition to behavioral and environmental factors that play a role in exercise adherence and adoption. We also explore strategies and principles that, although not commonly utilized in an obesity/chronic disease treatment setting, may be applied and adapted to fit this model.

## 1. Introduction

### 1.1. Lifestyle Obesity Treatments: Potential Advantages of Exercise

Obesity prevalence in the U.S. continues to be a growing concern. Most recent Center for Disease Control (**CDC**) data demonstrates 23 states have obesity rates exceeding 35%, while no states reached a 35% prevalence in 2013 [1]. Although new anti-obesity medications (**AOMs**) are demonstrating promise, AOMs are not accessible for all individuals due to issues with cost, insurance coverage, side effects, contraindications, and the need for long-term therapy. Thus, lifestyle obesity interventions are still considered first-line treatments. Such lifestyle approaches to obesity treatment are centered on reducing energy intake **(EI**) and increasing physical activity (**PA**), typically producing modest (5–10%) weight loss over 3–6 months [2,3,4]. Energy-restricted diets have been employed for decades, with some individuals seeing great weight loss success; however, adherence to energy-restricted diets can be difficult to sustain beyond 1–4 months, and as a result, some individuals are unable to achieve clinically meaningful weight loss with this approach [5,6,7]. There are several reasons some individuals struggle with adherence to energy-restricted diets, including environmental determinants (lack of access to healthy foods or an abundance of non-healthy foods, social cues/expectations), or individual factors (stress, emotions, cravings, or general preferences) [7,8,9,10]. For some individuals, chronic food restriction may produce psychological consequences such as eating binges once food is available, dysphoria, and distractibility, all of which limit weight loss [11]. Even when weight loss is achieved, one-third to two-thirds of weight lost by energy restriction is typically regained in 1 year, and almost all lost weight is typically regained in 5 years for up to 90% of adults [12]. Such weight cycling can produce metabolic issues, including glucose and insulin dysregulation, and cardiac dysrhythmias [13,14,15]. These negative effects and difficulties are not universal, and strategies have been developed to promote healthy diet-induced weight loss, including counseling with a registered dietitian, support groups, individually tailored meal plans, and additional behavioral training [16]. However, for some people, these additional strategies may not be available or preferable, and thus, focusing on exercise as the primary treatment for obesity and related comorbidities may be more appealing. In fact, 63% of U.S. adults who attempt weight loss report engaging in exercise as their primary strategy [17]. An additional benefit of exercise-focused weight loss interventions is that almost all of the weight that is lost is fat mass, whereas 15–30% of weight loss from energy restriction interventions can be lean mass [18]. Thus, an equivalent weight loss produced by exercise could have greater health benefits than that produced by diet alone. However, the response variability to exercise is wide, and special considerations are needed when employing exercise as a weight loss and chronic disease prevention strategy.

### 1.2. Energy Compensation Limits Weight Loss in Response to Exercise

The notion that exercise is a health-promoting activity is well recognized across professional organizations, with numerous consensus statements focusing on the benefits of exercise for obesity treatment [19] and related comorbidities such as cancer [20], Type II diabetes [21], cardiovascular disease [22,23], chronic pain [24], and fatty liver disease [25]. Although improvements in these diseases can be obtained through exercise independent from weight loss, such benefits are substantially improved when clinically meaningful (≥5–10%) weight loss is also achieved [2]. Despite the universal recommendations for exercise in obesity and its comorbidities, exercise rarely results in the amount of weight loss expected based on the energy expended in the exercise program [26,27]. Well-designed and tightly controlled longitudinal trials have consistently demonstrated that weight loss from exercise alone is only 30–40% of that predicted based on the measured energy expenditure of exercise sessions [28,29,30,31,32,33]. The discrepancy between the amount of weight loss predicted from exercise-associated energy expenditure and the observed weight loss has been termed “energy compensation”, driven by various “compensatory mechanisms” that function to maintain energy homeostasis by either promoting greater EI or conserving energy (reducing total daily energy expenditure, **TDEE**) [34,35]. Such mechanisms are considered evolutionarily conserved, once serving as a desirable trait to conserve energy for vital bodily functions, such as reproduction, during times of food shortage [26,27,34]. These traits, however, are a detriment in today’s environment, resisting intended weight loss and weight loss maintenance. Mechanisms contributing to these compensatory increases in EI or reductions in TDEE are complex and vary among individuals, contributing to the response heterogeneity observed with exercise-induced weight loss [30,36]. Disagreement also exists among scientists, with some proposing that energy compensation can be negated by large doses of exercise (3000 kcal/week) [33], while others point to greater exercise doses resulting in larger compensatory responses [28]. Other large trials have reached contrasting conclusions regarding the compensatory responses elicited from aerobic exercise. For instance, the Midwest Exercise Trial 2 (MET-2) demonstrated that “compensators” (individuals with <5% weight loss) decreased non-exercise physical activity (**NEPA**), whereas “non-compensators” (individuals with ≥5% weight loss) increased NEPA across a 10-month aerobic exercise intervention [37]. This is in contrast to the more recent E-MECHANIC trial where changes in EI (measured via doubly labeled water) increased 865 kcal/week when exercising to expend ~1760 kcal/week. At the same time, resting energy expenditure only slightly (non-significantly) increased by 135 kcal/week with virtually no changes in NEPA [38]. It is thus apparent that more research into the compensatory responses to exercise, mechanisms controlling these responses, sources of individual variability, and strategies to ameliorate energy compensation are needed if we are to understand how to optimize and personalize exercise for the treatment of obesity and chronic diseases.

### 1.3. Enhancing Effectiveness of Exercise in the Treatment of Obesity and Chronic Diseases

Exercise is commonly defined as “activity requiring physical effort, carried out to sustain or improve health and fitness.” In this way, not all activities requiring physical effort are considered “exercise,” but if the purpose is to improve/sustain health and fitness, the possibilities of what could be considered exercise are extensive [39]. Exercise scientists can thus modify many variables such as intensity (measured via heart rate or % of max effort), mode (aerobic to resistance exercise), frequency (sessions per week), duration of exercise sessions or duration of an intervention, or timing of exercise. The numerous combinations of these parameters and others give us the ability to create a wide variety of exercise treatments that produce substantially different physiological adaptations and require different considerations in terms of implementation into practice guidelines [40]. The purpose of this narrative review is not to outline how exercise is beneficial for obesity and related comorbidities, rather, we will delve into how exercise can be structured and implemented to improve its utility among individuals with obesity and chronic diseases. In this way, we can divide the “how to make exercise more effective” question into two aspects: (1) what physiological variables related to the exercise prescription can be targeted to make it a more effective weight loss treatment? and (2) How can we make exercise recommendations more translatable into practice, i.e., improve exercise adherence and adoption? In the sections that follow, we review several studies that may be applied to an obesity and chronic disease treatment perspective. Many of these trials and concepts reviewed have not been applied to these disease states; thus, an important future research focus should be testing these targets and concepts through clinical trials utilizing specifically participants with obesity or these certain conditions.

## 2. Physiological Targets to Improve Efficacy of Exercise in Promoting Improvements in Obesity and Associate Chronic Disease Outcomes

### 2.1. Considering the Physiological State in Which Exercise Is Performed: Post-Absorbative vs. Post-Prandial

The post-absorbative state (most commonly referred to as the fasted state) presents 6–12 h after a meal and is characterized by low levels of available carbohydrates (blood glucose, muscle glycogen) and insulin [41]. This is in contrast to the post-prandial state (most commonly referred to as the fed state), which is the hours immediately after eating. The lack of available carbohydrates promotes a shift in metabolic processes to more readily oxidize free fatty acids (**FFAs**) for energy, while low levels of blood insulin further enable lipolysis to ensue [42]. The presence and activity of many hormones are also very different in these opposing states, including cortisol, catecholamines, leptin, and ghrelin, issuing very different metabolic responses to environmental stimuli [43]. With the greater propensity to oxidize FFA in the fasted state, there has been considerable attention on fasted exercise training to promote FFA oxidation and improve oxidative capacity [44,45]. This has been demonstrated in acute exercise, where FFA oxidation was upregulated during and after fasted exercise compared to post-prandial (aka, fed state) exercise, leading to many theorizing that fasted exercise training would be more beneficial for weight management [46,47,48,49,50]. A recent review has even concluded that acute fasted exercise can create a larger acute energy deficit and thus could be a powerful weight loss intervention [51].

Although acute studies are helpful in understanding the underlying physiology and formulating hypotheses, longitudinal trials are needed to evaluate the utility of fast exercise training in promoting weight loss and improving chronic conditions associated with obesity. To our knowledge, only four short-term trials (4–6 weeks) have evaluated the efficacy of fasted exercise compared to exercise in the post-prandial state in a longitudinal nature, with three of them demonstrating significant improvements in a variety of FFA oxidative markers [47,48,49,50] (Table 1). Importantly, only one of these studies employed a dose of exercise aligned with current guidelines for weight management, and all of the studies were too short in duration to demonstrate clinically significant changes in weight loss or other markers of chronic disease. An additional study included in Table 1 aimed to quantify EI and EE for two days after an acute bout of exercise (75-min run) performed either in the fasted or post-prandial state [52]. This trial, similar to the 4–6 week interventions, did not detect a difference in EI or TDEE in the days after fasted exercise compared to fed exercise. Longer term trials (6+ months) are thus needed to determine if greater fat loss can ensue from fasted exercise training. If a significantly greater fat mass loss when exercising in the fasted state compared to an identical exercise prescription performed in the fed state can be demonstrated, this would represent an innovative method that may be employed to improve the weight loss response to exercise.

### 2.2. Considering the Time-of-Day Exercise Is Performed

The time of day that exercise is performed may impact weight loss and energy balance regulation [53]. Two observational studies have found preliminary evidence that morning physical activity is associated with lower body weight and lower body mass index (**BMI**) [54,55]. Similarly, Creasy et al. found that successful weight loss maintainers (i.e., individuals maintaining a weight loss of >30 lbs. for >1 year) engaged in 2-to-3 fold more moderate-to-vigorous physical activity (**MVPA**) in the morning (within 3 h of waking) compared to controls with and without obesity [56]. In a secondary analysis of the Midwest Exercise Trial 2, Willis et al. examined the effect of time of day of exercise on weight loss and energy balance [57]. Participants were categorized based on the time of day in which they completed the majority of their exercise sessions as follows: morning exercisers: >50% of sessions completed between 7:00 and 11:59 am; (*n* = 21, 70% of exercise sessions completed in the morning) and evening exercisers: >50% of sessions completed between 3:00 and 7:00 p.m.; (*n* = 25, 66% of exercise sessions completed in the evening). Morning exercisers lost significantly more weight than evening exercisers at 10 months (−7.2 ± 1.2% vs. −2.1 ± 1.0%). Interestingly, there were no differences between morning and evening exercisers in baseline characteristics, exercise adherence, or exercise energy expenditure (528 ± 105 vs. 490 ± 103 kcal/session, respectively). The differences in weight loss appeared to result from differential changes in EI and non-exercise expenditure (i.e., TDEE not associated with exercise training). Morning exercisers exhibited slight decreases in EI, while evening exercisers increased EI (−63 ± 444 vs. 121 ± 484 kcal/d, non-significant). In addition, morning exercisers exhibited slight increases in non-exercise expenditure compared to evening exercisers (28 ± 446 vs. −105 ± 510 kcal/d, non-significant). However, these results could have been affected by confounders as participants self-selected exercise times. In addition, that study only included young adults (18–30 years), which limits the generalizability of the study.

Only a few short-term (≤12 week) prospective studies have examined the effects of exercise timing on changes in body weight or fat mass in adults with overweight or obesity. Alizadeh et al. randomized women to 6 weeks of supervised morning or afternoon aerobic exercise [58]. The exercise dose was modest (90 min/wk). In a completers’ analysis, body weight significantly decreased in morning exercisers compared to afternoon exercisers (−1.6 vs. −0.3 kg). This difference in weight loss appeared to be due to changes in EI as self-reported EI tended to decrease more in morning exercisers compared to afternoon exercisers. Arciero et al. also found that women who completed 12 weeks of exercise in the morning lost more fat mass (−1.0 ± 0.2 kg) compared evening exercisers (−0.3 ± 0.2 kg) [59]. However, no differences in fat mass loss were observed for men. In a pilot and feasibility study, Creasy et al. found that individuals randomized to 2000 kcal/wk of morning aerobic exercise training lost −0.9 ± 2.8 kg, and evening exercisers lost −1.4 ± 2.3 kg with no significant differences between groups [60]. Two additional recent trials also found that 12 weeks of morning and evening exercise resulted in no significant differences in weight loss [61,62]. In contrast, Di Blasio et al. found that postmenopausal women participating in a 12-week walking program lost more fat mass if they exercised in the evening compared to the morning (−1.7 kg vs. −0.2 kg) [63]. Mancilla et al. also found that evening exercise led to greater reductions in fat mass compared to morning exercise after 12 weeks of training [64]. Thus, these short-term studies focused on the effect of exercise time of day on weight and fat mass have mixed results.

The lack of consistent findings may be due, in part, to the limitations of these prior studies and different exercise training variables between studies. For example, these studies have recruited heterogeneous populations, and it is possible that age, body size, sex, and other participant characteristics influence the effect of exercise time of day on weight loss. In addition, the duration of the exercise interventions (≤12 weeks) may have been too short to produce clinically meaningful weight loss for the dose employed [29,30]. Further, objective measures of compensatory mechanisms that affect energy balance regulation were not measured; thus, the effect of morning versus evening exercise on compensatory mechanisms remains unknown. Finally, some of these studies used exercise prescriptions that do not align with current exercise guidelines for weight loss [65]. Given the methodological limitations of these prior studies, there is a need for fully powered, randomized trials of longer durations and with an adequate dose of exercise to determine the effect of exercise timing on body composition [53].

### 2.3. Considering When to Initiate Exercise: Before or After Energy-Restricted Weight Loss

Most behavioral weight loss programs recommend concurrently decreasing EI and increasing physical activity/exercise. However, many individuals with overweight or obesity are unable to achieve and sustain high levels of exercise with this simultaneous approach [66,67,68,69]. An alternative strategy, which has not been widely considered, is to deliver diet and exercise interventions sequentially. Delaying the start of an exercise intervention until after an initial period of diet-induced weight loss could result in enhanced exercise adherence and improved long-term weight loss because (1) perceived enjoyment of exercise may be greater at a lower body weight [70], (2) risk of exercise-related injuries may be reduced after weight loss [71]. (3) Joint pain may be reduced, and thus exercise tolerance may be greater after weight loss [72], and (4) focusing on one behavioral change at a time may lead to greater long-term adherence to both diet and exercise behaviors [73,74].

However, two studies have suggested that the timing of exercise initiation does not impact weight loss. A trial by Goodpaster et al. enrolled 130 adults with class II obesity or higher (BMI  >  35 kg/m^2^) in a 1-year intensive lifestyle intervention consisting of diet and physical activity. Participants were randomized to either initial physical activity (diet and physical activity for the entire 12 months) or delayed physical activity (identical dietary intervention but with physical activity delayed for 6 months). Both groups achieved significant weight loss at 12 months (initial: 12.1 kg (95% CI, 10.0–14.2) vs. delayed: 9.9 kg (95% CI, 8.0–11.7), and there were no significant differences between groups in weight loss, waist circumference, blood pressure or insulin resistance [75]. Similarly, a study by Catenacci et al. enrolled 170 adults with overweight or obesity (BMI 27–42 kg/m^2^) into an 18-month behavioral weight loss program consisting of a reduced-energy diet, exercise, and group-based support [76]. Participants were randomized to either a standard group which received a supervised exercise program (progressing to 300 min/wk of moderate-intensity aerobic exercise) during months 0 to 6, or a sequential group, which was asked to refrain from changing exercise during months 0 to 6 and received the supervised exercise program during months 7 to 12. On completion of supervised exercise, both groups were instructed to continue 300 min/wk of moderate-intensity exercise for the study duration. At 18 months, both groups lost weight (standard: −6.9  ±  1.2 kg; sequential: −7.9  ±  1.2 kg), and there were no differences between groups in changes in weight, fat mass, lean mass, physical activity, or attrition. Combined, these studies suggest that both immediate and delayed exercise initiation within a behavioral weight loss program resulted in clinically meaningful weight loss and improvements in health across a range of BMI; thus, the timing of exercise initiation can be personalized based on patient preference.

### 2.4. Considering Mode of Exercise

Aerobic exercise is characterized by its extended and continuous duration (e.g., 10–40 min) at a low to moderate intensity with large muscle groups that challenge the delivery of oxygen to the active muscles [77]. This type of exercise has been at the forefront of clinical recommendations since Dr. Kenneth Cooper’s 1968 book “Aerobics,” which set the stage for exercise physiology research and integration into clinical practice as a health-promoting activity [78]. Since this early work, aerobic exercise has consistently demonstrated profound effects on various markers of chronic disease [79,80,81,82]. Chronic physiological adaptations to aerobic exercise include greater blood and stroke volume, greater cardiac output and perfusion, lower resting heart rate, improved ejection fraction, improved fatty-acid oxidative capacities, greater metabolic efficiency, and many others [83]. Common aerobic exercises, such as brisk walking, running, and cycling, are relatively simple to implement and require little specialized knowledge, helping make aerobic exercise the most common form of exercise both in terms of public health engagement and clinical recommendations. Also, walking is the most preferred mode of exercise for people with obesity [10].

Recent years have seen an increase in research focusing on strength/weight training exercises (also referred to as resistance training) involving the use of high-resistance machines or other external weights with specific movements limited to a few repetitions (generally less than 20) to reach or approach muscular exhaustion [83]. If considering a continuum where physical activities are either highly oxidative (low intensity, long duration) or highly glycolytic (short bursts, high intensity), resistance and aerobic exercise would occupy the opposite ends of this spectrum. Thus, the metabolic and physiological adaptations to resistance training have the potential to vary greatly from those of aerobic exercise. Despite this, more literature is demonstrating resistance training can have similar improvements in areas such as lowering blood pressure, improving insulin sensitivity and blood lipid profiles, decreasing the cardiovascular demands to exercise, and improving functional capacity [84]. Resistance training has demonstrated promising effects on health benefits for treating diseases such as cardiovascular disease [85,86], cancer [87], diabetes [88], and liver disease [89]. Resistance training may be a more appropriate exercise mode for individuals with factors such as obesity, arthritis, low back pain, and physical disabilities that make continuous aerobic activity difficult. By using machines that provide external resistance with controlled movements, even those confined to a wheelchair or a walker can perform some types of resistance training [88]. However, certain barriers present with resistance exercise training, such as the specialized knowledge and equipment compared to aerobic exercise training that may dissuade some individuals from resistance training. Supervision by a qualified professional and proper program design has been deemed the key elements of an effective resistance training program [84,90]. If these barriers can be overcome, resistance exercise can serve as an effective treatment for chronic diseases and weight loss for some individuals.

### 2.5. Considering the Intensity of Exercise Sessions

Exercise intensity can broadly be defined as the level of difficulty or exertion during exercise. Exercise intensity can be quantified (e.g., % heart rate max, % maximal oxygen uptake, % 1-repetition max, rating of perceived exertion) and qualified (e.g., light, moderate, vigorous) in dozens of ways, often dependent on the mode of exercise (e.g., aerobic or resistance). For aerobic exercise, the American College of Sports Medicine provides the following intensity categories based on percent of maximum heart rate (%HRmax): very light < 57%, light 57–63%, moderate 64–76%, vigorous 77–95% and maximal ≥ 96% [91]. Traditionally, moderate-intensity aerobic exercise has been the most studied, but the last 20 years have seen a shift to studying higher-intensity exercise. This is likely because, in general, exercising at a greater intensity is associated with greater health benefits [92]. In terms of comparing moderate to high intensity, multiple meta-analyses have found high intensity to be superior in terms of improving fat mass loss and body composition [93,94] and cardiorespiratory fitness [93,94,95]. Even in analyses that show comparable health effects, high-intensity exercise may take up to 40% less time [96]. This is vital, as time is often reported as a major barrier to exercise in those with obesity [10]. While potentially superior for certain outcomes, moderate-intensity aerobic exercise is still a viable option to improve waist circumference and body fat, especially when performed at least 150 min per week [97]. Therefore, the choice of whether to perform moderate or high-intensity exercise should likely be person-dependent and consider contextual factors such as time, motivation, resources, and preference.

A great deal of attention has been spent on studying the health benefits of moderate-to-high-intensity exercise, but perhaps equally important is light-intensity physical activity. During waking hours, the majority of time is not spent performing moderate-to-high intensity exercise rather, it is performing daily activities or engaging in sedentary behavior. While attaining physical activity recommendations should continue to be emphasized, we must acknowledge that despite efforts over the past three decades, the proportion of adults meeting physical activity guidelines remains low (24.2% of U.S. adults in 2020) [98]. Performing higher volumes of physical activity is associated with reduced mortality risk, regardless of intensity [99]. Thus, incorporating more movement and reducing sedentary time should be a public health focus. An example of this may be the inclusion of “exercise snacks” or short bouts of purposeful exercise (e.g., stair climbing) interspersed throughout the day [100]. A scoping review in adults and older adults found that exercise snacks were feasible and may be associated with reduced all-cause mortality and reduced risk of major cardiovascular events [101]. While research on exercise snacks is in its infancy, they may serve as a viable alternative to exercise occurring in a more formal setting such as a fitness center.

For resistance exercise, intensity is measured by load or the amount of weight used in an exercise, typically in relation to a person’s one repetition maximum (**1-RM**) [102]. The intensity of resistance exercise is dependent on whether the goal is to increase muscle strength, hypertrophy, power, and/or endurance. For example, if maximizing muscle strength is the goal, resistance exercises should be performed at ≥60% 1-RM in individuals unaccustomed to resistance exercise and ≥80% 1-RM for those with experience [102,103]. While resistance training alone benefits body composition, a recent meta-analysis compared seven types of exercise (including aerobic only, resistance only) at various intensities and found that for improving lean body mass and decreasing abdominal fat, combining high-intensity aerobic (i.e., >75% heart rate max) with high-intensity resistance (>75% 1-RM) training was best [104]. While this multi-modal training does not always lead to outcomes that are superior to a single modality [105], they may be more time-efficient in terms of meeting physical activity guidelines, which have both an aerobic and strength component.

High-intensity functional training (**HIFT**) is a form of exercise that combines aerobic and resistance exercises, with an emphasis on movements that replicate real-life task demands such as carrying groceries or climbing stairs [106]. HIFT has been shown to improve strength, power, speed, endurance, and agility in healthy males [107] and, when compared to traditional circuit training, improve body composition in females [108]. Compared to a combined aerobic plus resistance training program performed at a moderate intensity, HIFT participants spent less time exercising but reported similar levels of enjoyment and greater intentions to continue to exercise [109]. HIFT may be even more effective at improving body composition when combined with time-restricted eating. In a three-armed study comparing time-restricted eating, HIFT, and a combined HIFT plus time-restricted eating, those in the combined group observed greater decreases in fat mass compared to the other groups [110]. Given the range of benefits following HIFT participation, it’s not surprising that as of 2023, HIFT was the sixth most popular fitness trend in the U.S. With the time-efficient nature of combining aerobic and resistance exercise, HIFT is another option to combat the common barrier of lack of time to exercise.

## 3. Adoption of Exercise–Translating from Structured Interventions to Real-World Behavior Change

### 3.1. Applying the Socioecological Model

An overarching premise to consider is that an exercise intervention that cannot be maintained will not be a very effective treatment for overweight/obesity and chronic diseases. Thus, it is important to understand factors that can promote adherence to an exercise intervention. The socioecological model provides a comprehensive framework for understanding how various levels of influence—from individual factors to broader environmental and policy-related factors—affect health behaviors like exercise adherence [111]. This framework emphasizes that behavior is shaped not only by personal choices but also by the social, community, and environmental contexts in which individuals live. Applying the Socioecological Model to exercise adherence is increasingly recommended for promoting long-term weight loss and improving chronic disease outcomes, as it recognizes that interventions must target multiple levels of influence to be effective. However, improving exercise adherence across all levels of the socioecological model presents unique and multifaceted challenges, requiring a focus on both individual factors and environmental determinants [112].

Key personal characteristics, such as motivation, self-efficacy, exercise history, social support, and stage of change/habit, are critical determinants of exercise engagement [10,113,114]. Research indicates that women are often motivated by social factors, while men are typically driven by perceived health benefits [112,115]. Despite these differences, exercise initiation and adoption are often cyclical, with engagement fluctuating in response to varying motivational inputs [116]. Social support is a key social determinant of exercise adherence and can significantly influence activity levels, with family, coworkers, and community support all playing potentially meaningful roles in improving exercise engagement [112,117]. The relationship between exercise and social support is dynamic, evolving over time as individuals progress through different stages of health behavior change [115]. Time constraints, a common barrier to regular exercise, can be mitigated by establishing a “time hierarchy” that prioritizes exercise despite stressful and competing life demands [118]. While interventions that target social support and time management have shown promise, further research is needed to understand their impact on behavior change in specific populations, particularly those in rural and food-insecure communities, where barriers to both time and access to exercise facilities may be more pronounced [119,120,121]. This underscores the need for interventions that address both personal and environmental factors to foster long-term engagement and develop effective strategies that support exercise adherence in underserved populations [122].

### 3.2. Theory, Targeting, and Tailoring Behavioral Interventions

Interventions based on theoretical frameworks and practical strategies have effectively increased exercise adherence [123]. One such framework is social cognitive theory (**SCT**), which promotes self-management behaviors, such as adopting a healthy lifestyle, through self-regulating cognitive processes [124]. Evidence supports the application of SCT in PA interventions, showing small-to-moderate effect sizes for PA improvements [125,126]. Key SCT domains—social support, self-efficacy, self-regulation, and outcome expectations—have been identified as significant mechanisms of action for increasing PA, particularly among rural populations [127,128]. Interventions that incorporate community engagement strategies have been shown to improve health behavior outcomes, including social support and self-efficacy, in underserved populations [129]. These findings highlight the importance of addressing both individual and environmental determinants to foster long-term exercise adherence. More advanced models combine SCT with dual-process theories, which incorporate constructs like environmental cues, affective evaluations, and automatic behavioral impulses [130,131].

Building on the need for targeted theoretical frameworks, the selection and combination of effective behavior change techniques (**BCTs**) are crucial for enhancing the efficacy of interventions aimed at improving exercise adherence [132]. Michie et al. emphasize that BCTs, when strategically chosen and tailored to the specific needs of the target population, can significantly improve health-related behaviors, including exercise [133]. Through tailoring and taking into account personal preferences, needs, and environmental context, interventions can increase the likelihood of successfully increasing exercise adherence [134]. Short message service (**SMS**) or text messaging has increasingly demonstrated promise as a delivery vehicle for improving exercise behaviors by providing personalized support and encouragement [135]. Additionally, gamification—the use of game elements in non-game contexts—is an encouraging modality to promote and change exercise behaviors [136]. Further research is needed to understand how the use of SMS-based BCT and/or gamified interventions can address the unique challenges faced by diverse populations, including individuals of varying ages, socioeconomic backgrounds, and regions, ensuring that interventions are tailored to meet the specific needs and barriers of each group [137].

Human-centered design (**HCD**), or design thinking, offers another promising approach by focusing on the behaviors, needs, and experiences of individuals, ensuring that interventions are both relevant and sustainable [138]. HCD’s iterative process of empathy, prototyping, and testing facilitates the development of tailored solutions that are not only feasible but also resonate with the target population [139]. Applying HCD principles to exercise adherence and adoption enables the creation of strategies that address varying levels of readiness for change, resource access, and personal barriers to physical activity, making the interventions more adaptable and effective for diverse populations [140]. Using HCD principles, interventions can be better tailored to the specific needs and preferences of the target population.

### 3.3. Motivation and Exercise Adherence

Research has identified several motivational factors that influence exercise adherence [114,141]. Intrinsic motivation, where individuals engage in exercise for the enjoyment, challenge, or personal satisfaction it provides, plays a critical role in long-term adherence [142]. Younger adults, in particular, who find exercise personally fulfilling are more likely to maintain a regular routine [143]. In contrast, extrinsic motivation, driven by external rewards like praise or appearance concerns, can undermine long-term adherence, especially when those rewards are no longer present [144].

The intrinsic motivation to exercise has been objectively quantified in the literature by assessing one’s reinforcing value of exercise, a measure of how much an individual wants to work for exercise [145,146,147,148], and subjectively, with survey-based measures of wanting or desiring to move and exercise [149]. Research by Flack et al. has demonstrated the reinforcing value of a specific mode of exercise, such as resistance or aerobic training, is a stronger predictor of participation than simply liking the activity [150], while in a subsequent trial, Flack and colleagues found that exercise reinforcement was associated with more frequent physical activity and meeting activity guidelines [151]. These findings demonstrate the reinforcing value of exercise as a primary determinant in the choice to exercise and, thus, underscores the importance of such intrinsic motivation in exercise adherence [152].

The aforementioned motivational constructs have typically been considered from a trait perspective, whereas motivation is viewed as relatively stable and difficult to change [153]. Motivation to move may even be a classic drive, similar to hunger or the need for rest. In other words, a lack of movement results in internal tension to ambulate, which is only resolved when the behavior is consummated. This may be altered for those with obesity, such that they have weak urges to move but very strong urges to be sedentary [35,153]. Such a prospect necessitates that motivation also be considered as a construct that changes frequently, just like hunger or appetite (or appetence, which is the fluctuating desire for physical activity). Indeed, attention has turned to the idea of motivational states for physical activity, exercise, and sedentary behaviors, including sleep [154]. In short, motivation for behavior can change quickly, even in just a few minutes. This has been exhibited with changes observed in the desire to move with a variety of activities, including maximal exercise or periods of rest [149], short bursts of activity [155], exercise training sessions [156], or even just talking about exercise [157]. Stronger exercise intensities have a greater impact on motivation state [158]. There is also a circadian effect, whereas for most people, the desire to move peaks after 3 pm [159,160]. Motivation states also vary between apparently healthy and clinical populations [156]. Motivation states have typically been described as weak (desires, wants) to strong (urges, cravings), often with a feeling of tension, which might be positive (groove) or negative (urges to move when unwanted); thus, they have been described as “affectively-charged motivation states” (**ACMS**).

Importantly, ACMS during exercise has been shown to predict salient exercise behavioral factors, such as stage of change [149] and PA intention [159]. They also predict affect/emotion in subsequent training sessions [161]. This is important as affective responses to exercise predict future exercise engagement [162]. Motivation and affect-based models, such as the affect health behavior framework [163], include ACMS and expand the idea to highlight the role of dread for exercise, ostensibly common for those with obesity who have had poor exercise experiences. Dread and other aversions for exercise, or movement more broadly, have been contrasted against wants/desires/cravings in the WANT model (Wants and Aversions for Neuromuscular Tasks), which blends approach and avoidance motivation for physically active and inactive behaviors [154]. There is limited data linking affectively-charged motivations to actual behavior, with the exception of Crosley-Lyons [160], who found those with higher ACMS in the morning exhibited about 28 extra minutes of moderate-to-vigorous exercise later in the day. Further work is ongoing [164], and there is a need for data in populations with obesity, though data in limited student populations indicates that motivation states are not related to BMI [149].

### 3.4. Interventions to Increase Exercise Motivation

The reinforcing aspects of exercise are products of the central dopamine reward system, eliciting a dopamine release to promote exercise as a reinforcing behavior [165]. Similar to other behaviors that elicit a dopamine release and are thus considered highly reinforcing (drugs of abuse, alcohol), certain genetic phenotypes are associated with dopamine release to influence exercise reinforcement and behavior [166,167,168]. This indicates that certain individuals are more prone to realizing exercise as a reinforcing activity and, thus, interventions aimed at increasing the reinforcing value of exercise may only be applicable to individuals with certain genotypes, which complicates research in this area. Increasing the reinforcing value of behavior is accomplished through “incentive sensitization”, which occurs when repeated exposures increase the salience of a stimulus within the environment [169]. This results in neuroadaptations that sensitize the dopamine reward system, thus increasing the reinforcing value of the behavior [170,171].

Several interventions aimed at increasing exercise reinforcement through the process of incentive sensitization have been conducted with mixed results. One trial demonstrated that low levels of exercise exposures (3 days/week for 6 weeks, either 150 or 300 kcal/session) reduced the reinforcing value of sedentary activities but did not significantly increase exercise reinforcement [172]. A follow-up trial demonstrated that exercise reinforcement can be significantly increased when exercise exposures are of greater volume (5 sessions/week, 300 or 600 kcal/session for 12 weeks) [173], which is in line with the dose-response relationship in developing drug abuse reinforcement [174]. This finding was replicated in a similar high-dose intervention but extended, determining increases in exercise reinforcement predicted an increased engagement in habitual physical activity after the structured exercise intervention (via accelerometry) [175]. Taken together, these results indicate that a high-dose exercise intervention can provide the necessary exposures to exercise to sensitize the dopamine reward system to the reinforcing properties of exercise (incentive sensitization). Importantly, this process of incentive sensitization can then, as hypothesized, be an effective way to increase exercise engagement.

Interventions focusing on motivation and affect, specifically, affectively-charged motivation states, are currently being tested and show promise [176]. As one example, environmental cues might prompt a person to exercise, and a person sensitive to this type of stimulus might demonstrate a rise in desire or urge to move. Other environmental factors, such as daylight, temperature, and music, also likely impact ACMS [157]. Also to consider is the timing of exercise in relation to meals, as feeding results in lower motivation to move initially, with a subsequent increase with additional fuel, and then a decline after an hour [159]. Interventions could be tailored to the time of day when motivation states are their highest [159,160]. Increasing the rewarding value of exercise for this population, perhaps by considering individual preferences, theoretically should increase the liking and thus the wanting of movement [154]. Particularly with vigorous exercise, the desire to move reduces, and the desire to be sedentary increases [154,156,158]. With this in mind, it may be ideal to implement exercise that is of moderate intensity to avoid quick movement satiation and maintain exercise appetence. Thus, motivation states may also be an important automatic compensatory response to exercise, which might limit the effectiveness of some exercise interventions [35]. With all of this in mind, tracking motivation states may help to optimize the timing of exercise sessions, taking advantage of motivation when it is high, and intervening in motivation itself when it is low, possibly as part of a just-in-time adaptive intervention [177] which may be a novel strategy to intervene via mobile technology. Recent work by Dunton et al. and Barrows et al. provide additional theoretical and practical applications of modifying motivation states to improve adherence [176,178].

### 3.5. Addressing Exercise Adherence and Maintenance at the Systems Level

The majority of research on promoting exercise has been focused on the individual level (i.e., a person’s motivations, barriers, facilitators). To promote long-term exercise, however, there is a recommendation for interventions to target not just individuals but additional levels of the social-ecological model, including social and built environments [179,180], as well as considering determinants of implementation within these levels [181]. In other words, while the question, “how do we facilitate individuals to exercise?” still needs to be addressed, equally important is the question, “how do we support workplaces, communities, and policies that facilitate individuals to exercise?” The latter question is addressed in the field of implementation science, which studies how to best help people and places implement the intervention, which, in this case, is exercise [182].

Rather than designing, implementing, and refining a new exercise program to combat obesity, which may take up to 17 years to become part of routine practice [183], a more practical approach is to capitalize on an existing evidence-based intervention. In this way, we move from studying effectiveness (i.e., does it work in real-world settings) to implementation (i.e., how do we make it work in real-life settings). The National Institutes of Health have resources that evaluate the scientific rigor of exercise and nutritional interventions and include implementation guides that can facilitate the uptake of these evidence-based interventions across various settings (i.e., work, school, community). By studying these existing, ready-to-implement interventions, we may be able to reduce the 17-year gap between research and practice and have a larger reach and impact on human health.

### 3.6. Worksite Health Promotion: Employer’s Role in Supporting Exercise Adherence

Employers play a key role in promoting exercise adherence by integrating physical activity into workplace health programs and policies [184]. Tailoring exercise prescriptions based on job-related activity levels and providing opportunities for movement, such as flexible schedules and active transportation incentives, can help employees incorporate exercise into their routines [185]. Additionally, offering fitness memberships, physical activity assessments, and referrals to community resources can further support adherence [186]. Notably, the Physical Activity Alliance’s CEO Pledge encourages organizational leaders to model healthy behaviors, creating a culture that normalizes physical activity [187]. By adopting these strategies, employers can promote sustained employee health, reduce healthcare costs, and improve long-term exercise adherence [188].

## 4. Conclusions

The obesity crisis in the U.S. necessitates interventions that overcome adherence barriers and compensatory physiological responses undermining traditional lifestyle approaches. Exercise is vital for weight management and chronic disease prevention, but its effectiveness is constrained by compensatory mechanisms resulting in increased energy intake and/or reduced expenditure, requiring strategic optimization. The present review does not provide concrete answers on how to make exercise more effective for obesity and chronic disease treatment, rather, we present many concepts that may be considered in future trials and interventions that may aim to improve obesity and chronic disease outcomes through exercise. Modifying exercise variables—such as timing (e.g., AM vs. PM sessions), mode (aerobic vs. resistance), and intensity—can enhance fat oxidation while mitigating compensatory behaviors. Understanding motivational processes, including incentive sensitization, basic drive, and motivation states, holds promise in strengthening intrinsic motivation and long-term adherence by leveraging neurobiological reward pathways. Behavioral strategies, including gamification and personalized feedback, are targets for interventions. A socioecological framework is essential, addressing individual motivation, social support, workplace policies, and environmental accessibility to create sustainable physical activity ecosystems. Employers and policymakers play critical roles in scaling solutions through workplace wellness programs, community infrastructure, and culturally relevant campaigns to normalize exercise as a daily priority.

There is currently a great deal of conflicting results and limitations in the research surrounding each of these concepts, thus making it difficult to draw conclusions. For instance, we have ample evidence that fasted exercise training can improve fatty-acid oxidation, although no trials have been undertaken in a clinical population or in patients with obesity at the necessary duration to elicit changes in body weight or disease state. Conflicting results are present when looking at the time of day exercise is performed, where strong evidence obtained from secondary data analysis of large exercise interventions point to morning exercise best-supporting weight loss, while randomized trials manipulating exercise time of day have not shown this. Out of the topics we have covered, it does appear resistance training and high-intensity exercise have several advantages when it comes to weight loss, although this type of training may not be suitable for certain clinical populations, and more research is still needed. On the exercise adherence and behavioral side, it does appear that individuals can be sensitized to the reinforcing effects of exercise to increase exercise adherence. Similarly, several trials elucidating the role of ACMS on exercise behavior have demonstrated environmental cues (daylight, temperature, music) and food intake can influence one’s motivation to exercise and thus play an important role in adherence.

It is important to note that many of the studies reviewed herein were not carried out in a population with obesity or focused on treating a clinical condition. This limitation further stresses the importance of future research, which must prioritize prospective studies to clarify optimal exercise timing, dosing, and motivational and compensatory mechanisms in this population. Technology must be leveraged for real-time adherence tracking, metabolic feedback, and motivational messaging. A multidisciplinary approach—integrating physiology, psychology, as well as environmental design—is indispensable to transforming exercise from a set of population-wide guidelines into a personalized, evidence-based intervention. Overall, it is apparent that overcoming obesity and improving chronic disease outcomes require systemic, flexibly adaptive strategies that empower individuals through science-driven, context-sensitive solutions to adopt and sustain active lifestyles.

## Figures and Tables

**Table 1 nutrients-17-01997-t001:** Trials comparing fasted vs. fed exercise training on weight change and muscle biochemistry.

Study	Intervention	Weight Loss	Fat/Carbohydrate Oxidation Markers
Schoenfeld, et al. 2014 [47]	Aerobic exercise, 60 min, 3×/week plus dietary restriction.	No difference between fasted and fed groups.	None.
Van Proeyen, et al. 2010 [48]	Aerobic exercise, 30–60 min, 4× week, 6 weeks plus hyper-energetic diet.	Fasted exercise attenuated weight gain.	Fasted exercise increased FFA oxidative markers (AMPK, CD36, CPT1).
Van Proeyen, et al. 2011 [50]	Aerobic exercise, 60–90 min, 4×/week, 6 weeks plus isoenergetic diets.	None.	Fasted exercise increased lipid breakdown, maximal fat oxidation, increased FFA oxidative markers (CS, B-HAD).
Gillen, et al. 2013 [49]	Interval training aerobic exercise, 20 min, 3×/week, 6 weeks.	No difference between fasted and fed groups.	Fasted exercise increased FFA oxidative markers (CS, B-HAD).
Blannin, et al. 2024 [52]	75-min run, energy intake and expenditure assessed for 2 days after.	No difference in energy intake or expenditure between fasted and fed groups.	No difference in interstitial glucose between fasted and fed groups.

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
