# Peer review of "Exploring Strategies to Promote Exercise as a Viable Obesity and Chronic Disease Treatment"

_nutrients, 2025, doi:10.3390/nu17121997_

Round 1

Reviewer 1 Report

Comments and Suggestions for Authors

Many thanks to the editor for the opportunity to revise this manuscript.

In this narrative review, Falck et al. aim to explore how physical exercise could be structured to promote weight loss and improve obesity management. Please find below my comments.

The authors state that lifestyle interventions, including physical exercise and caloric restriction, represent the first line treatment for obesity. However, when introducing the role of physical exercise (section 1.1), the role of nutrition appears to be framed somewhat negatively. I believe this may not fully reflect the current evidence, as nutrition and physical exercise form a fundamental and synergistic pair. For lifestyle interventions to be truly effective, these two components must go hand in hand. For instance, the authors suggest that exercise primarily leads to fat mass loss, whereas caloric restriction also results in lean mass loss. This statement may not be entirely correct. In fact, if exercise is not supported by an adequate nutritional plan, it may also lead to lean mass loss. Moreover, to optimize the physiological adaptations to training, it is essential that any exercise program be complemented by appropriate nutritional support. I would encourage the authors to revise this section accordingly.

In the various sections of the manuscript, the authors provide a general overview of each topic, followed by conclusions drawn from the available evidence (for example, the need for long term studies to determine whether fasted training leads to greater fat mass loss compared to fed-state training, or the heterogenous results concerning the optimal time of day for exercise). However, many of the studies discussed involve athletes, healthy individuals, or overweight participants, while relatively few involve individuals with obesity. Drawing conclusions based on populations that differ from the target of this review may be speculative. I would encourage the authors to revise the various sections by prioritizing studies that specifically include individuals with obesity, where available. To enhance clarity, the inclusion of a summary table might also be helpful.

The inclusion of Parkinson’s disease and type 1 diabetes mellitus, which are not directly associated with obesity (as acknowledged by the authors), may rise some concerns regarding the coherence of the manuscript. Given that these conditions are addressed only briefly, their inclusion may detract from the central aim of the review.

The conclusion section should more clearly reflect the message of the review. As stated, the authors aim to investigate how physical exercise could be structured to enhance weight loss and how it could be implemented to improve long-term adherence. Therefore, it would be helpful for this section to briefly summarize the key findings from the literature discussed throughout the manuscript. Specifically, based on the available evidence (preferably from studies involving individuals with obesity) it would be valuable to highlight which exercise modalities, intensities, timing strategies, and motivational techniques appear most promising in promoting greater weight loss and long-term compliance.

Author Response

Comment 1: … when introducing the role of physical exercise (section 1.1), the role of nutrition appears to be framed somewhat negatively. I believe this may not fully reflect the current evidence, as nutrition and physical exercise form a fundamental and synergistic pair... I would encourage the authors to revise this section accordingly.

Response: Thank you for the comment and perspective. We agree that a sound nutrition plan is important and were not intentionally downplaying the role of nutrition. We were attempting to convey that weight loss via energy restriction has drawbacks while weight loss via exercise has certain benefits. To make this clear, we revised to ensure we are speaking of energy restriction as opposed to the more broad “dietary-induced weight loss” that we used in a few places.

Comment 2: … many of the studies discussed involve athletes, healthy individuals, or overweight participants, while relatively few involve individuals with obesity. Drawing conclusions based on populations that differ from the target of this review may be speculative. I would encourage the authors to revise the various sections by prioritizing studies that specifically include individuals with obesity

Response: Thank you for noticing this and bringing it to our attention. We do recognize that many of the studies cited did not specifically utilize participants with obesity or focus on obesity treatment, and thus `may not seem appropriate to draw conclusions for obesity treatment. Although, this was actually our intention- bringing together evidence that may be applied to a weight loss/obesity treatment perspective. Using only studies that featured participants with obesity would not have resulted in many studies reviewed. We therefore added, at the very end of section 1.3 the following: In the sections that follow, we review several studies that may be applied to an obesity-treatment perspective, although many of these trials and concepts reviewed have not been applied to this scenario. An important future research focus should be testing these targets and concepts though a clinical trial utilizing specifically participants with obesity.

Comment 3: The inclusion of Parkinson’s disease and type 1 diabetes mellitus, which are not directly associated with obesity (as acknowledged by the authors), may rise some concerns regarding the coherence of the manuscript. Given that these conditions are addressed only briefly, their inclusion may detract from the central aim of the review.

Response: Thank you for this comment, we agree these conditions are a bit ancillary and have removed these sections from the manuscript.

Comment 4: The conclusion section should more clearly reflect the message of the review. As stated, the authors aim to investigate how physical exercise could be structured to enhance weight loss and how it could be implemented to improve long-term adherence… it would be valuable to highlight which exercise modalities, intensities, timing strategies, and motivational techniques appear most promising in promoting greater weight loss and long-term compliance.

Response: We appreciate this comment and recognize that many readers would expect concrete answers to the questions posed in a review article. However, this was not a meta-analysis or a systematic review that focused on answering a question, rather, we proposed concepts that may be applicable to obesity and chronic disease treatment. Many of the concepts reviewed did not yield a concrete answer. For example, the notion of fasted vs. fed exercise training there has yet to be a trial of sufficient length or dose to determine if one form of exercise training better supports obesity / chronic disease treatment, but there is evidence that it may. The purpose of this review was to bring these concepts to attention and detail how they may be applied to obesity / chronic disease treatment. To make this clearer to the reader, we incorporated the following into the conclusion: The present review does not provide concrete answers on how to make exercise more effective for obesity and chronic disease treatment, rather we present many concepts that may be considered in future trials and interventions that may aim to improve obesity and chronic disease outcomes though exercise.

Reviewer 2 Report

Comments and Suggestions for Authors

Dear Authors,

Congratulation for an interesting topic that permit many reflections about obesity treatment with exercise.

For us the objectives of you study are clear: “For obesity and related chronic diseases treatment, question into two aspects: 1) what physiological variables related to the exercise prescription can be targeted to make it a more effective weight loss treatment? and 2) How can we make exercise recommendations more translatable into practice, i.e., improve exercise adherence and adoption?” We do not understand the link with: “Type I diabetes and Parkinson’s Disease and how they can garnish immense benefits from exercise.” For us this is another review and should be removed of the present revision. They are two separates topics with exercise impact as common point, but the focus is obesity and not chronic diseases.

Probably a better title for your study should be:  How to make exercise more effective for obesity and associated chronic diseases treatment? Please consider.

Line 37, When the abbreviation CDC is used for the first time, the full term should be written out before it. The same for NEPA in line 99.

Lines 131 and 132, we suggest: 2. Physiological Targets to Improve Efficacy of Exercise in Promoting Weight Loss and Improvements in Obesity and Associate Chronic Disease Outcomes

Lines 133 to 169 (point 2.1.), Please specify if you are referring to long or short term fasting. Muscle glycogen depletion probably is more related to long term fasting before exercise. Please address this in the text and eventually the table.

Line 166 you write: “These trials indicate that longer term trials (6+ months) are needed” witch one of the presented studies had this duration? Or on the basis of what argumentation 6+ months are recommended? Please address this.

Line 187 you write: “Morning exercisers lost significantly more weight than evening exercisers at 10 months” What are the invoked reasons? What about the methodologies used in the studies and the possible influence of confounding factors? Please address this.

In 2.2., it is important to consider if in the presented studies the training was comparable in the two situations.

Line 218 to 219 and 222 to 223, please add a reference for each affirmation.

Line 225, we suggest “adequate and equivalent dose”

In 2.4. and 2.5., you do not present result about the impact of strength training on weight loss. You present effect on lean body mass but what about fat mass with strength training, or HIIT or combination aerobic and strength. Please address this and compare with continuous aerobic training.

Point 4 for us is to suppress. We suggest you change for a practical guideline to improve obesity treatment with exercise. And making suggestions for futures studies.

Author Response

Comment 1: …We do not understand the link with: “Type I diabetes and Parkinson’s Disease and how they can garnish immense benefits from exercise.” For us this is another review and should be removed of the present revision.

Response: Thank you for this comment, we agree these conditions are a bit ancillary and have removed these sections from the manuscript.

Comment 2: Probably a better title for your study should be:  How to make exercise more effective for obesity and associated chronic diseases treatment? Please consider.

Response: Thank you for this suggestion, after a great deal of deliberation, we have decided to keep the original title. Although your suggestion would be great in most cases, we feel that we are not making recommendations or reviewing evidence to make recommendations as to how to make exercise more effective in this narrative review. We are rather outlining some concepts that future research may consider. Thus we feel a title starting with “how to make exercise….” would be too strong of a statement for this manuscript and not what we were intending on conveying.

Comment 3: Line 37, When the abbreviation CDC is used for the first time, the full term should be written out before it. The same for NEPA in line 99.

Response: thank you for noticing this, we have made these changes.

Comment 4: Lines 131 and 132, we suggest: 2. Physiological Targets to Improve Efficacy of Exercise in Promoting Weight Loss and Improvements in Obesity and Associate Chronic Disease Outcomes

Response: Thank you for this suggestion, we have used your idea for a section title except we took out the “weight loss” mention as this seemed redundant to improvements in obesity. It now reads: Physiological Targets to Improve Efficacy of Exercise in Promoting Improvements in Obesity and Associate Chronic Disease Outcomes

Comment 5: Lines 133 to 169 (point 2.1.), Please specify if you are referring to long or short term fasting. Muscle glycogen depletion probably is more related to long term fasting before exercise. Please address this in the text and eventually the table.

Response: We appreciate this comment, and although we are not sure how this reviewer defines short vs long term fasting, we have added the following to the first sentence of this section to make it clear: The post-absorbative state (most commonly referred to as the fasted state) presents 6-12 hours after a meal and is characterized by low levels of available carbohydrate (blood glucose, muscle glycogen) and insulin [45].

Comment 6: Line 166 you write: “These trials indicate that longer term trials (6+ months) are needed” witch one of the presented studies had this duration? Or on the basis of what argumentation 6+ months are recommended? Please address this.

Response: thank you for this comment, as we realize this may not have been clear. There were no studies of this duration, which is why we called for a need for them. We have thus taken the “These trials indicate that” out of this sentence to avoid this confusion.

Comment 7: Line 187 you write: “Morning exercisers lost significantly more weight than evening exercisers at 10 months” What are the invoked reasons? What about the methodologies used in the studies and the possible influence of confounding factors? Please address this.

Response: Thank you for this question, we have provided more insight by adding the following at the end of this section: The differences in weight loss appeared to result from differential changes in EI and non-exercise expenditure (i.e., TDEE not associated with exercise training). Morning exercisers exhibited slight decreases in EI, while evening exercisers increased EI (-63 ± 444 vs 121 ± 484 kcal/d, n.s.). In addition, morning exercisers exhibited slight increases in non-exercise expenditure compared to evening exercisers (28 ± 446 vs -105 ± 510 kcal/d, n.s.).

Comment 8: In 2.2., it is important to consider if in the presented studies the training was comparable in the two situations.

Response: Thank you for this comment, we certainly agree with this notion. The studies presented in this section compared morning vs evening exercisers in the same study (either through secondary analysis or as designed). Since these groups were in the same study, they received the same exercise stimulus. We do agree that different exercise variables across studies would make it difficult to compare, thus we added the following to the last this paragraph : The lack of consistent findings may be due, in part, to the limitations of these prior studies and different exercise training variables between trials  

Comment 9: Line 218 to 219 and 222 to 223, please add a reference for each affirmation.

Response: Thank you for pointing these out, we have made one slight change in wording and added the necessary references.

Comment 10: Line 225, we suggest “adequate and equivalent dose”

Response: Thank you for this suggesting, we agree and have made this change.

Comment 11: In 2.4. and 2.5., you do not present result about the impact of strength training on weight loss. You present effect on lean body mass but what about fat mass with strength training, or HIIT or combination aerobic and strength. Please address this and compare with continuous aerobic training.

Response: Thank you for pointing this out, we have added sentences referencing the following studies: 97, 98,108,114, all of which compared the effects of either resistance exercise or high-intensity exercise to moderate intensity aerobic exercise on weight loss/body composition changes/ fat mass loss.

Comment 12: Point 4 for us is to suppress. We suggest you change for a practical guideline to improve obesity treatment with exercise. And making suggestions for futures studies.

Response: thank you for this suggestion, we included a sentence in the conclusion where we state what future research must prioritize in order to set personalized guidelines for exercise and obesity/chronic disease treatment.

Round 2

Reviewer 1 Report

Comments and Suggestions for Authors

I thank the authors for their responses. However, I still have some concerns. Please, find below my comments.

Comment to response 1: I appreciate the authors’ effort to address this point and refine the terminology. However, I believe that simply replacing “dietary-induced weight loss” with “energy restriction” may not fully clarify the message. Upon re-reading the introduction, the impression remains that exercise is portrayed as more favorable than caloric restriction. Caloric restriction and physical exercise should be complementary strategies, each supporting the effectiveness of the other. Moreover, if appropriately planned, caloric restriction does not necessarily cause adverse effects, such as psychological consequences or weight regain (as stated by the authors). I encourage a further revision of this section to better reflect this synergy.

Comment to response 2: I thank the authors for their clarification. However, if the intent is to explore strategies that may be applicable to obesity treatment, this aim should be more explicitly stated, beginning with the title, which in its current form may be somewhat misleading. A revised manuscript structure could further support this objective. For instance, organizing the content to first present findings supported by evidence in individuals with obesity, followed by concepts that are not yet tested in this population but with potential applicability, would enhance transparency and help readers better contextualize the relevance of each approach. Additionally, the limited number of studies conducted in individuals with obesity, and the extrapolation of data from other populations, could be acknowledged as a limitation in an appropriate section of the manuscript.

Comment to response 4: While the goal of this review is not to provide definitive answers, even in a narrative review it may be helpful to briefly summarize which approaches, based on the current evidence, appear most promising. This would offer the reader a clearer takeaway message and help differentiate between concepts that remain theoretical and those with emerging support.

Author Response

Comment to response 1: I appreciate the authors’ effort to address this point and refine the terminology. However, I believe that simply replacing “dietary-induced weight loss” with “energy restriction” may not fully clarify the message. Upon re-reading the introduction, the impression remains that exercise is portrayed as more favorable than caloric restriction. Caloric restriction and physical exercise should be complementary strategies, each supporting the effectiveness of the other. Moreover, if appropriately planned, caloric restriction does not necessarily cause adverse effects, such as psychological consequences or weight regain (as stated by the authors). I encourage a further revision of this section to better reflect this synergy.

Response to Comment to response 1: We would like to thank the reviewer for this perspective, and it is true that we are trying to promote exercise as an obesity and chronic disease treatment that may, in some cases, be more favorable than energy restriction. We do agree that some people may have great success with energy restriction, use it in conjunction with exercise, and do not experience the adverse effects as we mention. However, we are attempting to present exercise as an alternative avenue that may be more favorable for others. We have thus revised to soften our stance, stating that many people may have success with energy restriction, but at the same time, others may struggle with this method and exercise may be a more effective strategy. 

Comment to response 2: I thank the authors for their clarification. However, if the intent is to explore strategies that may be applicable to obesity treatment, this aim should be more explicitly stated, beginning with the title, which in its current form may be somewhat misleading. A revised manuscript structure could further support this objective. For instance, organizing the content to first present findings supported by evidence in individuals with obesity, followed by concepts that are not yet tested in this population but with potential applicability, would enhance transparency and help readers better contextualize the relevance of each approach. Additionally, the limited number of studies conducted in individuals with obesity, and the extrapolation of data from other populations, could be acknowledged as a limitation in an appropriate section of the manuscript.

Response to Comment to response 2: We appreciate the reviewer’s recommendations and agree that this aim needs to be more explicit to avoid confusion. We have thus changed the title to read “Exploring Strategies to Promote Exercise as a Viable Obesity and Chronic Disease Treatment”.  We have also extended the abstract to call reader’s attention to the idea that “We also explore strategies and principles that, although not commonly utilized in an obesity / chronic disease treatment setting, may be applied and adapted to fit this model”.  We also include a similar statement at the end of section 1.3 to set the stage for the sections that follow. We do appreciate the idea of breaking out the trials that utilized an obese or clinical population; however, many of these sections have no such trials (for example, 2.1, fed vs fasted). To be more transparent, we did note, in the conclusion, the limitation that most of the studies reviewed were not specific to obesity and/or chronic disease and called for more research in this area.

Comment to response 4: While the goal of this review is not to provide definitive answers, even in a narrative review it may be helpful to briefly summarize which approaches, based on the current evidence, appear most promising. This would offer the reader a clearer takeaway message and help differentiate between concepts that remain theoretical and those with emerging support.

Response to Comment to response 4: We appreciate this suggestion and have added an entire paragraph to the conclusion, outlining the concepts where research appears to be most robust while recognizing that it is difficult to make recommendations due to the conflicting results or lack of long-term trials in obese/clinical populations for most of the topics we have covered.

Reviewer 2 Report

Comments and Suggestions for Authors

Dear Authors,

thank you in considering all our observations.

We wish you success with this publication

Author Response

We would like to thank the reviewers for their time in reviewing this manuscript.